# Temporal assortment of cooperators in the spatial prisoner's dilemma

Tim Johnson [1,2 ✉] & Oleg Smirnov [3]

We study a spatial, one-shot prisoner's dilemma (PD) model in which selection operates on both an organism's behavioral strategy (cooperate or defect) and its decision of when to implement that strategy, which we depict as an organism's choice of one point in time, out of a set of discrete time slots, at which to carry out its PD strategy. Results indicate selection for cooperators across various time slots and parameter settings, including parameter settings in which cooperation would not evolve in an exclusively spatial model—as in work investigating exogenously imposed temporal networks. Moreover, in the presence of time slots, cooperators' portion of the population grows even under different combinations of spatial structure, transition rules, and update dynamics, though rates of cooperator fixation decline under pairwise comparison and synchronous updating. These findings indicate that, under certain evolutionary processes, merely existing in time and space promotes the evolution of cooperation.

[1] Atkinson Graduate School of Management, Willamette University, Salem, OR 97301, USA. [2] Center for Governance and Public Policy Research, Willamette University, Salem, OR 97301, USA. [3] Department of Political Science, Stony Brook University, Stony Brook, NY 11794, USA. ✉email: tjohnson@willamette.edu

Cooperation not only occurs at specific *locations*, but, also, it takes place at particular *points in time*. Indeed, whenever organisms interact in the flesh, cooperation necessarily occurs at both a spatial location and temporal moment.

For instance, at night[1,2], spinner dolphins collectively herd prey, creating food aggregations that exceed the prey densities encountered by lone foragers[3]. After amassing their prey, the encircling herders enter the accumulation in pairs and take turns feeding[3]. Each pair feeds for roughly the same duration, thus resisting the opportunity to enjoy a disproportionate individual gain that would diminish others' benefits[3]. Also, only participants in the herding appear to partake in the feeding[3], even though, conceivably, non-herders could lurk nearby and attempt to consume the accumulated prey. This herding behavior amounts to cooperation in a prisoner's dilemma: the herders create a benefit, $b$, at a cost, $c$, that free riders could consume without paying a cost. The behavior's subtleties, however, obscure a seemingly mundane feature of it—namely, that it occurs at one point in time (a particular moment at night), as opposed to other points in time.

The temporal concentration of cooperation also occurs in other species. Consider humans engaged in illicit market transactions that occur beyond the scope of institutions that support cooperative trade. These traders aim to acquire a benefit, $b$, by exchanging a resource that they part with at personal cost, $c$. However, they do best when their trade partner delivers $b$, while they transfer poor-quality goods or inadequate currency that reduce or nullify $c$. To avoid such cheating, illegal barterers schedule the *time* of transactions deliberately, or express preferences for some hours over others[4]. Cooperation, these activities imply, succeeds more frequently at certain times.

Albeit possible that proximate environmental factors (e.g., illuminance) explain the time of behavior in these examples, the examples also raise the possibility that the time of behavioral implementation—by itself—might act as a mechanism that allows cooperators to separate themselves from defectors. Thus, we ask —does selection on organisms' decisions of *when* to interact influence the evolution of cooperation?

Here we answer that question via a computer simulation in which organisms adopt strategies consisting of a tuple: (i) a choice in PD play (cooperate or defect) and (ii) the time at which to implement that behavior. Time slots, $t$, in the model vary from 1 to 10. When $t = 1$, the model amounts to a spatial model in which all spatially adjacent organisms interact with each other. For $t > 1$, organisms at the same spatial location might not interact. Organisms only interact with spatially adjacent organisms who are *also* in their same time slot—that is, who implement their behavior at the same time. If two agents in the same time slot both choose to cooperate in the PD, they receive $b-c$, where $b$ ranges from 1 to 10 and $c$ is fixed at 1; mutual defection earns zero, whereas free-riding earns $b$ and exploited cooperators suffer $-c$. We run this simulation for 10000 generations, repeating the simulation for 7 runs at each parameter setting. We, furthermore, perform our simulation on two separate spatial structures—a regular lattice and a small-world network. Organisms in the simulated population are arrayed across the nodes of those graphs, with the population size, $N$, set at 100, 225, and 400. In the small-world network model, we vary $k$, the average number of edges per organisms, from 2 to 4. In models involving each spatial structure, we also vary the transition rule that governs strategy change; we study the model under both death-birth fitness proportional selection (hereafter, "fitness proportional selection") and pairwise comparison[5,6]. Moreover, we examine the effect of alternative update dynamics by studying the model under asynchronous versus synchronous updating[5,7,8]. Via these simulation methods, our investigation adds to the literature on the spatial PD[9–13], as well as to recent efforts[14,15] to model the role of intragenerational time in social evolution.

Specifically, we build on the spatial games framework first proposed in Chapter 8 of Axelrod's *The Evolution of Cooperation*[16] and revolutionized by Nowak and May[9]. Consistent with this paradigm[7,12,17], we ignore cognitively complex strategies that, for instance, identify and cooperate with kin[18–20], prior cooperators[21,22], phenotypic doppelgangers[23–26], or economic equals[27–29]. Instead, we study a population of organisms, positioned on graphs, who adopt zero-intelligence[30,31] strategies that always cooperate or defect in one-shot PD games.

Nowak and May[9–11] revealed that playing those games on a grid and updating strategies based on neighborhood comparisons yielded clusters of cooperators that persist in the population alongside defectors. Subsequent research explored how modeling subtleties influenced the success of cooperation in this framework[7,32] and it extended the framework's reach into the study of heterogeneous networks[7,33–35], including the development of universal rules to characterize the conditions in which cooperation evolves in any spatial structure[36]. We augment such models by considering how the addition of a temporal dimension to spatial PD models influences social evolution. To do so, we create the possibility for organisms at any node of the spatial structure to choose one out of a fixed set of time slots to implement their behavioral strategy; selection then operates on this choice behavior.

Adding these time slots might appear redundant with existing models. If one regards space and time as interchangeable (for instance, interpreting a three-dimensional model as depicting a two-dimensional space with time as the third dimension), then models of the spatial prisoner's dilemma[11] would seem already to have examined the case we consider. Yet such an interpretation requires a researcher to assume that temporally adjacent organisms interact like spatially adjacent organisms. We do not assume so. In our model, only agents in the same time slot—not adjacent time slots—interact. This distinguishes our model from multidimensional, spatial PD models and it formalizes what we believe is a plausible notion: organisms that stand elbow-to-elbow at the same time likely interact, whereas those that wake at sunset don't interact with those who are fast asleep at that same time, even if they are physically proximate.

Our model might also be mistaken with spatial PD models studying movement. Regardless of whether movement occurs because of stochastic drives[37–39], pre-planned patterns of travel[40], tendencies toward cohesive collective movement[41,42], or the attributes of an organism's current[43–46] or prospective location[47–51], movement constitutes a change in spatial location that occurs across time. Thus, two moving agents that interact implicitly do so at the same moment. In such models, however, the instant of social interaction does not vary. Movement occurs, interaction transpires, and this game play happens either at one particular time point or in a repeated fashion such that organisms interact with others at all available time points. Time of behavioral implementation, therefore, remains homogenous among organisms in extant models involving movement. Alternatively, we allow organisms in the population to vary when, within a generation, they interact. (Time would play a more salient role in a model of movement in which agents move perpetually within a generation, bumping occasionally into other organisms and commencing a PD game. We know of no such model presently, though we propose it in the discussion.)

By allowing varied times for social interaction, our model also differs from the compelling conceptualization of temporal assortment by Aktipis[52], which focuses on the delayed effects of an organism's behavior on its fitness. A deeper connection exists, however, between our model and studies involving strategic

timing[15] and temporal networks[14]. Our model resembles research into strategic timing in that we also examine how cooperation is influenced by the choice of when, within a generation, organisms implement social behavior. Yet, unlike research into timing[15], payoffs in our model do not vary based on their relation to particular events (such as the provision of resources from a public good); instead, they vary based upon the strategies adopted by other organisms active in the same spatial location at the same point in time. Treating the choice of time at which to implement PD behavior as a part of an organism's strategy also differentiates our work from research into the effect of temporal networks on cooperation[14]. Research on temporal networks[14] depicts the time of PD activity as exogenous and studies how the addition of multiple, temporal network layers influences cooperation. Our study allows for the endogenous development of such networks by letting selection operate on organisms' choices of when to implement their behavioral strategy of cooperation or defection.

Finally, our investigation also dovetails with a recent model by Wang and colleagues that studies a finite population whose members are divided into groups to play the public goods game, but who are subject to individual-level selection across the population[53]. The group affiliations in this model function like the time slots in our model, as they determine with whom simulated individuals interact; then, selection occurs across the entire population, as in our investigation, not within groups. Our model primarily differs from this recent work because of its (i) inclusion of a spatial dimension, (ii) focus on the one-shot PD, and (iii) applicability to the study of time in social evolution. However, the choice of a discrete time point at which to implement a PD strategy bears important technical similarity to the choice of joining a group in which to play a public goods game, thus warranting future consideration of the implications of these related models.

In our study, we find that adding the opportunity for organisms to implement their behavioral strategy at a discrete time point leads to the evolution of cooperation under a wide variety of parameter settings. Furthermore, increasing the number of time slots within the range examined in our model boosts the likelihood of cooperator fixation. When fixation does not occur, simulation runs with more time slots exhibit larger shares of cooperators in their terminal generation. Also, across combinations of spatial structures, transition rules, and update dynamics, we find higher rates of cooperator fixation and growth in the presence of time slots.

## Results

**Lattice model**. In Fig. 1, we present a typical simulation run in which (a) the population interacts on a lattice, (b) fitness proportional selection governs strategy transitions, (c) updating occurs asynchronously, and (d) $b = 2$, $N = 225$, and $t$ is set, respectively, at 1, 5, and 10. When $t = 1$ (Fig. 1a, leftmost panel), defectors dominate the population as all agents pool into the only time slot (Fig. 1b, leftmost panel). When $t = 5$, cooperators come to dominate the population (Fig. 1a, middle panel) and selection favors behavioral implementation at multiple time slots (Fig. 1b, middle panel). When we increase the number of time slots to its maximal value, $t = 10$, cooperators again grow to fixation (Fig. 1a, rightmost panel) and the population evolves to implement that strategy at 5 different time slots (Fig. 1b, rightmost panel).

Figure 2 displays the growth and decline of strategies across generations for the example runs displayed in Fig. 1. With $t = 1$, the proportion of defectors in the population immediately overtakes that of cooperators (Fig. 2, leftmost panels). However, with $t = 5$, cooperators resist the growth of defectors as selection favors cooperators at time slots in which they are populous

(Fig. 2, center panels). Close inspection of the lower-middle panel of Fig. 2 indicates selection for defectors in a time slot when the respective number of cooperators and defectors in that time slot are modest; however, defectors appear to extinguish themselves as they grow more prolific, driving selection for cooperators in time slots where those prosocial agents already thrive. Temporal assortment of cooperators, in sum, drives the population toward higher levels of cooperation.

As in these example runs, results across simulation runs that use the same transition rule and update dynamic show selection for cooperators in the presence of time slots. When $t = 1$ (all agents interact at the same time), cooperators reach fixation in 31.4% of all runs, yet the addition of one additional time slot ($t = 2$) results in a slight majority of runs (52.9%) reaching fixation of cooperators. The percent of runs resulting in the fixation of cooperators grows with the addition of each time slot, reaching 80% of all runs at $t = 5$ and 86.2% of runs at $t = 10$. Indeed, cooperation reaches fixation in 77.6% of all runs when $t \geq 2$. However, the increasing likelihood of cooperator fixation with growth in the number of time slots clearly has theoretical limits as astutely pointed out by an anonymous reviewer of this article. Were $t \rightarrow N$ or exceed it, the population would become sparsely distributed across time slots, leading inevitably to an asocial state in which cooperation would fail. In the discussion section, we propose analytic approaches to gain insight into such possibilities, though we preclude such phenomena by our choice of parameters in the current model.

As evident in Fig. 3, the frequency of the population reaching fixation of cooperators also grows with the value of $b$. Only in one anomalous case do we find cooperation growing to fixation when $b = 1$. When $b \geq 2$, cooperation reaches fixation in 81.1% of all runs. At first glance, Fig. 3 seems to suggest that the findings violate Ohtsuki et al.'s rule for the evolution of cooperation on graphs, which stipulates that selection favors cooperation when the ratio of benefits to costs exceeds $k$, the average number of neighboring organisms[36]. In 54.8% of all runs, cooperators grow to fixation when $b/c \not> k$ (i.e. when $b \leq 4$, given $c = 1$ and $k = 4$) and multiple time slots are present ($t \geq 2$). This apparent violation of the canonical rule $b/c > k$, however, is superficial; as another anonymous reviewer adroitly recognized, organisms experience an *effective* $k$, which we label $k^*$, denoting the average neighbors at a given temporal-spatial location. Were organisms uniformly distributed across spatial locations and time slots, $k^*$ would equal $k/t$. However, random seeding of time slots and the evolution of game play in those slots distributes the population unevenly; thus, $k^* = k/t$ constitutes the minimum value that parameter can take. This equation indicates that the addition of time slots has the effect of decreasing the value of $k$—a phenomenon observed in other recent research[14,54]. When $t > 1$, $k^*$ is less than $k$ and one would expect cooperation to evolve when $b/c > k^*$, thus explaining why *apparent* violations of the rule $b/c > k$ appear in the data.

The panels of Fig. 3 suggest that the value of $N$ also influences the success of cooperation. When $N = 225$, 79.0% of runs result in the fixation of cooperators. When $N = 400$, 55.9% of runs do so, whereas 84.1% result in the fixation of cooperators when $N = 100$. Does this pattern indicate the success of defectors or the slower growth of cooperators? First, when $N = 225$, 85.7% of all runs result in more than three-quarters of the population adopting the cooperator strategy at G = 10000. When $N = 400$, 84.4% of all runs end with more than three-quarters of the population adopting the cooperator strategy at $G = 10000$. Second, on average, when $N = 100$, cooperators reach fixation at generation, $G = 1709$; when $N = 225$, the average generation at which cooperators reach fixation is 3386; when $N = 400$, this average generation is 3502. Together, these pieces of evidence imply that cooperation still grows in larger populations, but it reaches

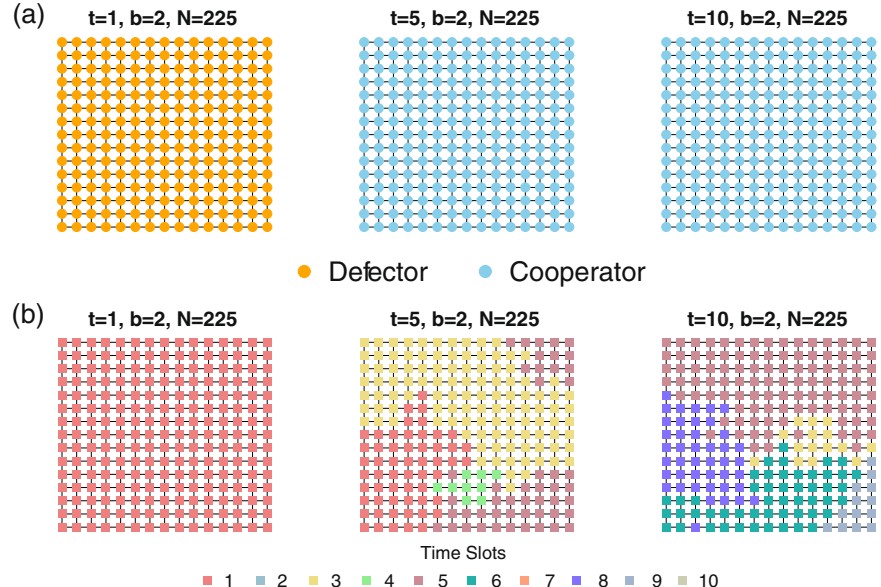

**Fig. 1 Evolution of behavioral strategies and time of implementation in an example run.** The figure displays the final distribution of behavioral strategies (**a**) and time slots in which behavior is implemented (**b**) in an example run of the simulation. The lattices in each column result from data generated in the same simulation run, with parameters set at those listed above each grid. Selection favors defectors when $t = 1$ (**a**, leftmost panel) with all agents pooled into the same time slot at G = 10000 (**b**, leftmost panel). When $t = 5$, cooperators come to dominate the population (**a**, middle panel) and selection leads to behavioral implementation at 4 time slots (**b**, middle panel). When $t = 10$: fixation of cooperators occurs (**a**, rightmost panel) and the population evolves to implement that strategy at 5 different time slots (**b**, rightmost panel).

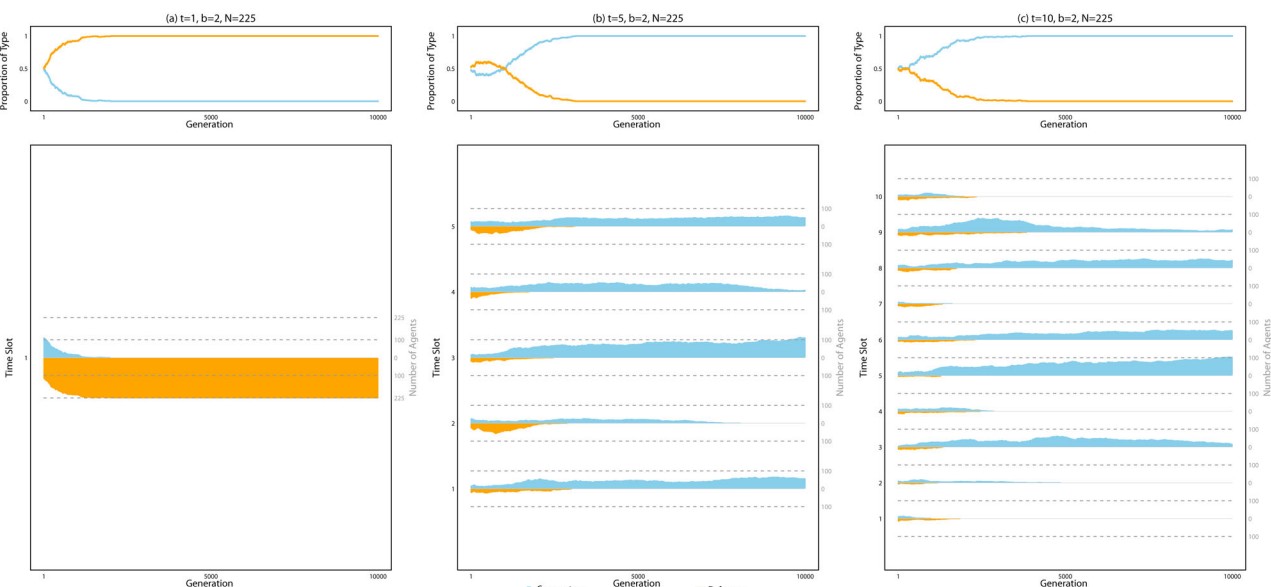

**Fig. 2 Selection for cooperation in the presence of multiple time slots in an example run of the simulation.** Panels display data from the same example simulation runs displayed in Fig. 1, with parameter settings appearing above the uppermost panel in each column. The upper row of panels displays the respective proportions of cooperators (light blue) and defectors (orange) in the population at each of the 10000 generations of the simulation run. The lower row of panels shows the number of cooperators and defectors in each time slot across generations, with time slots labeled on the left vertical axis and the right vertical axis measuring the number of cooperators and defectors at each time slot. Note that the vertical distance above 0 on the vertical axis measures the number of cooperators and the vertical distance below 0 (i.e., absolute value) measures the number of defectors. When $t = 5$ (middle panels) or $t = 10$ (right panels), selection favors cooperation and prosocial behavior evolves at multiple time slots.

fixation at a slower rate, thus leading to a greater frequency of runs in which cooperation cannot spread through the entire population by $G = 10000$, even though a plethora of agents adopt it.

**Small-world network model**. Analysis of selection on behavioral types and time of implementation in a small-world network with fitness proportional selection and asynchronous updating yields qualitatively similar results (see Fig. 4). Adding time slots to the model and allowing selection to operate on organisms' choices of which time slot to implement behavior increases the frequency of simulation runs resulting in the fixation of cooperators. When $t = 1$, 37.2% of all runs end with the fixation of cooperators; that percent increases to 56.7% when $t = 2$, grows to 76.9% when

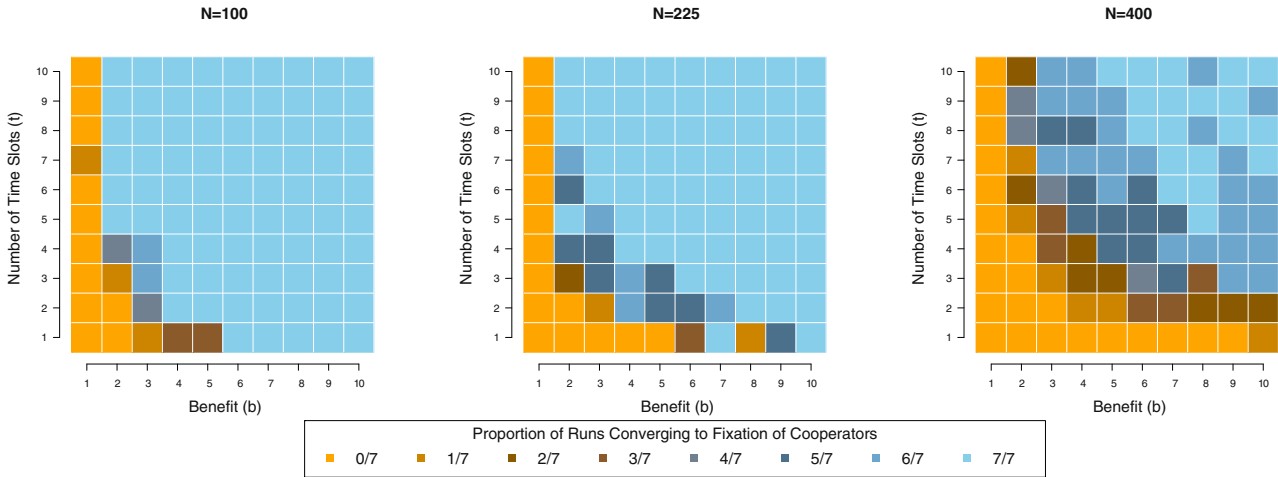

**Fig. 3 Cooperator fixation becomes more frequent across runs as benefit (*b*) and the number of time slots (*t*) increase in the lattice model.** We replicate our simulation for 7 runs at each parameter setting and display how the proportion of runs reaching fixation of cooperators varies by the values of *b* (horizontal axis) and *t* (vertical axis), displaying results for each value of *N* separately (as listed above each panel). Cooperation effectively never reaches fixation when *b* = 1, signifying the dearth of gains cooperators obtain even when interacting with each other in such situations. With *b* > 2, we observe populations in which cooperators grow to fixation; the prevalence of light blue tiles in the northeast corner of the grids indicates the increased success of cooperation when both *b* and *t* take large values within the model's parameter range. Notably, we observe less instances of cooperators reaching fixation as population size grows; as discussed in the text, this pattern relates to the speed of cooperators' growth as opposed to their ultimate success.

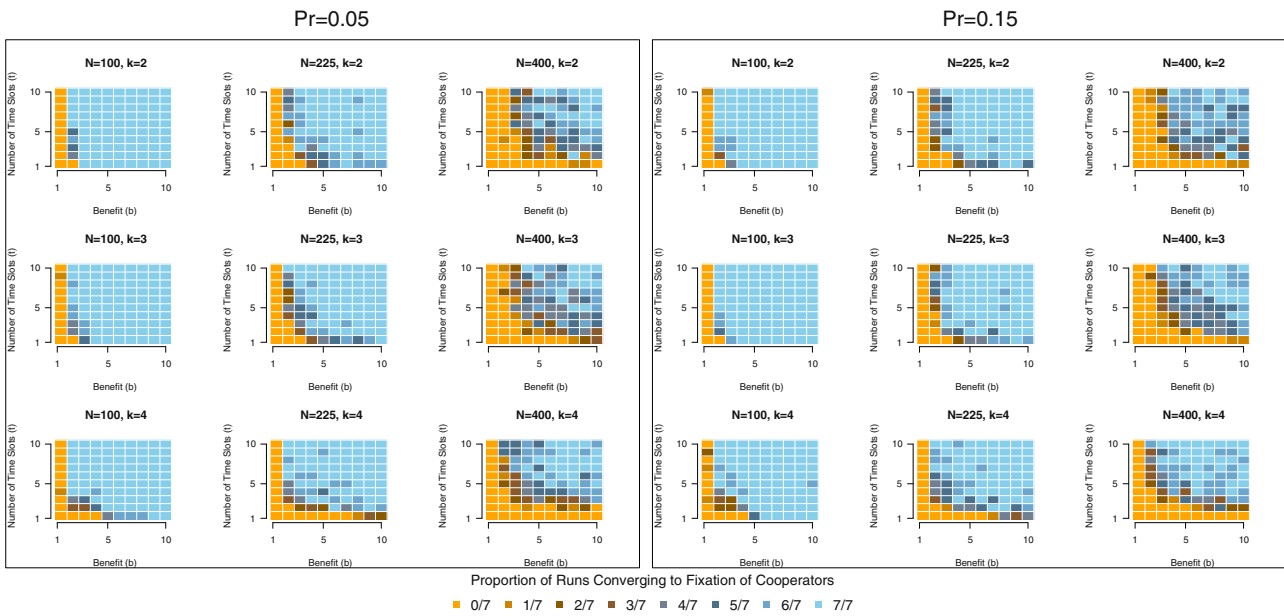

**Fig. 4 Cooperator fixation becomes more frequent across runs as benefit (*b*) and the number of time slots (*t*) increase in the small-world network model.** The panels display the frequency of cooperator fixation in simulation runs in which agents reside in a small-world network. Regardless of rewiring probability, *Pr*, we observe cooperator fixation more frequently as *b* and *t* take higher values.

*t* = 5, and peaks at 83.9% when *t* = 10. Likewise, as the value of *b* increases, the frequency of runs in which cooperators reach fixation increases; simulation runs ending with a population replete with cooperators occur extremely rarely when *b* = 1, but 45.5% of runs end with cooperator fixation when *b* = 2 and 91.2% of runs do when *b* = 10. As evident in Fig. 4, we again see that the population converges more slowly with larger values of *N*. Additionally, varying the probability of rewiring, *Pr*, does little to influence the simulation's results—as a comparison of the left- and right-half of Fig. 4 indicates. When *Pr* = 0.05, 71.7%% of all runs result in the fixation of cooperators; a comparable 72.6% of all runs result in the fixation of cooperators when *Pr* = 0.15. Accordingly, when we explore this model under alternative

transition rules and update processes (see below), we set the rewiring probability solely to *Pr* = 0.05 in order to limit simulation runtime to a manageable duration. Similarly, we find little evidence that rates of cooperator fixation vary across values of *k*: for *k* = 2, *k* = 3, and *k* = 4, we observe, respectively, 72.1%, 72.2%, and 72.0% of simulation runs reaching cooperator fixation.

**Sensitivity analysis.** The frequency of cooperator fixation, however, does vary in sensitivity analyses aimed at examining the consequences of altering the transition rule and update process of our simulation[5,6,8,55]. In those sensitivity analyses, we replicated our simulation under combinations of two different transition rules (fitness proportional selection and pairwise comparison)

**Table 1 Design of sensitivity analysis to study the effect of varying transition rules and update dynamics for the lattice model.**

|  |  | Update Dynamics | |
| --- | --- | --- | --- |
|  |  | Asynchronous | Synchronous |
| Transition rule | Death-birth fitness proportional selection | (a) | (b) |
|  | Pairwise comparison | (c) | (d) |

Table 1 presents an index denoting the combinations of transition rules and update dynamics studied in our sensitivity analysis of the lattice model. We use this index in Figs. 5 and 6 as a means of indicating the methods that underlie the results reported in those portions of this article.

**Table 2 Design of sensitivity analysis to study the effect of varying transition rules and update dynamics for the small-world network model.**

|  |  | Update Dynamics | |
| --- | --- | --- | --- |
|  |  | Asynchronous | Synchronous |
| Transition rule | Death-birth fitness proportional selection | (e) | (f) |
|  | Pairwise comparison | (g) | (h) |

Table 2 presents an index denoting the combinations of transition rules and update dynamics studied in our sensitivity analysis of the small-world network model. We use this index in Figs. 5 and 6 as a means of indicating the methods that underlie the results reported in those portions of this article.

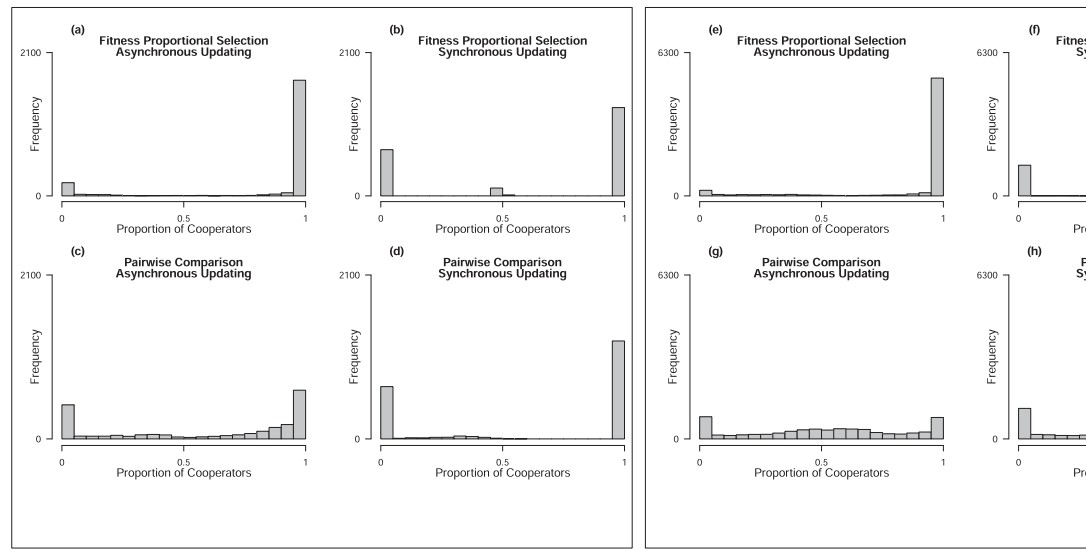
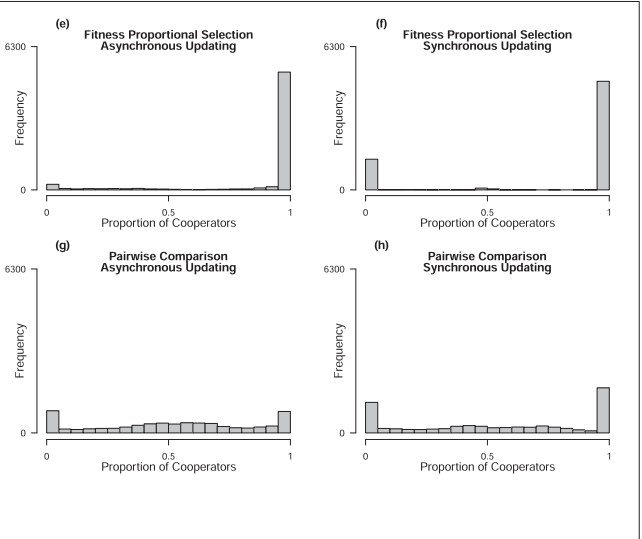

**Fig. 5 Final proportion of cooperators in the population under various combinations of transition rules and update dynamics.** Panels present the count of simulation runs with a given proportion of cooperators at $G = 10000$ under different transition rules and update dynamics. **a–d** Present data from models in which agents interacted on a lattice, while **e–h** present data from models in which agents interacted in a small-world network. The index presented in Table 1, as well as the main heading of each panel, distinguishes the combination of transition rule and update dynamics used in the simulation that generated data for a given panel. In each panel, we sort data into 5-unit bins of the horizontal axis, beginning at the minimal possible value (zero) and proceeding to the maximal possible value (unity); the gray bars then display the count of simulation runs (i.e., frequency) that ended with a proportion of cooperators resting within the range of a given bin.

and two separate update dynamics (synchronous and asynchronous updating), leading to a 2 × 2 research design for each spatial structure (see Tables 1 and 2). We treat the transition rule and update dynamics deployed in the models reported above (viz. fitness proportional selection and asynchronous updating) as a baseline condition.

In the lattice model, we found that about 73% of all runs in our baseline condition ended with the fixation of cooperators, approximately 8.1% of all runs ended with fixation of defectors, and 18.9% of all runs failed to converge (Fig. 5a). Under fitness proportional selection and synchronous updating (Fig. 5b), the

percent of runs in which cooperators reached fixation was fewer (61.6%), though it still exceeded the percent of runs resulting in defector fixation (32.2%), and only 6.1% of runs failed to converge. Failed convergence occurred more frequently when pairwise selection served as the model's transition rule; 63.8% of runs failed to converge when that transition rule was combined with asynchronous updating (Fig. 5c) and 14.3% of runs failed to converge when pairwise selection was combined with synchronous updating (Fig. 5d). When convergence succeeded, fixation of cooperators occurred more frequently than defector fixation in models with pairwise comparison. When updating was

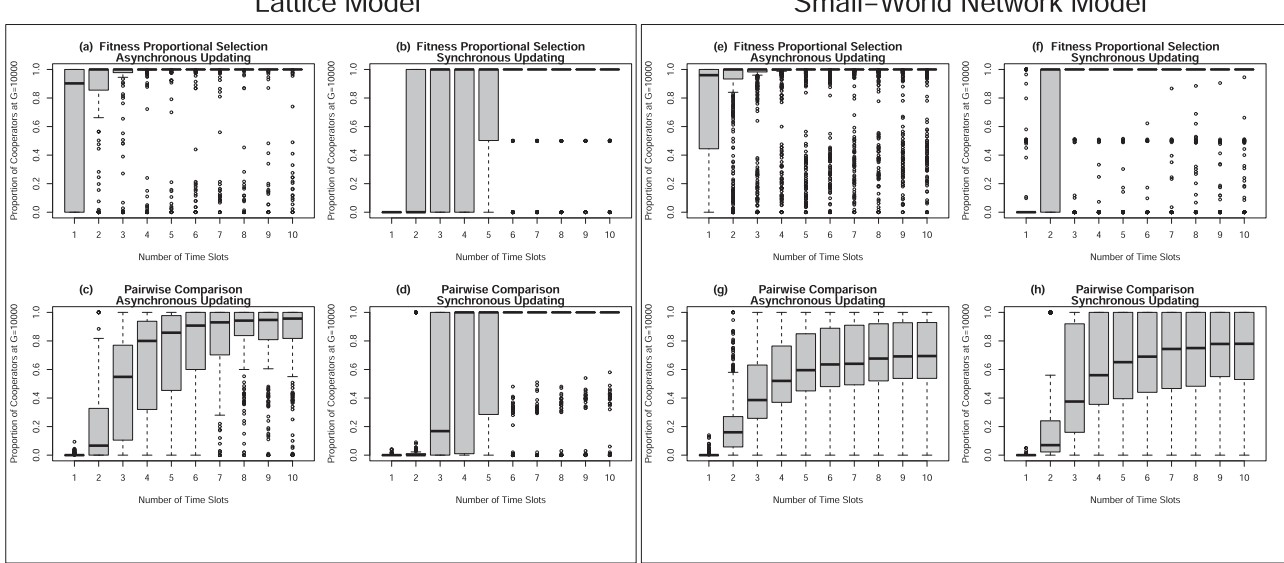

**Fig. 6 Varying the number of time slots influences the evolution of cooperation under combinations of transition rules and update dynamics.** Panels present box plots that display information about the distribution of cooperators' final population share across simulation runs, when data are separated by the number of time slots in those runs. **a–d** Present data from models in which agents interacted on a lattice, while **e–h** present data from models in which agents interacted in a small-world network. The index presented in Table 1, as well as the main heading of each panel, distinguishes the combination of transition rule and update dynamics used in simulations generating the data for a given panel. Heavy black lines denote the median value of the proportion of cooperators at $G = 10000$. The upper edge of a gray rectangle denotes the 75th percentile of the proportion of cooperators at $G = 10000$ and the lower edge of a gray rectangle denotes the 25th percentile, such that the full gray rectangle denotes the inter-quartile range (IQR). Points represent values that reside either a distance more than 1.5 times the length of the IQR above the 75th percentile or a distance more than 1.5 times the absolute length of the IQR below the 25th percentile. The short horizontal lines perpendicular to vertical dashed lines denote either the maximal value that rests less than 1.5 times the span of the IQR above the 75th percentile or the minimal value residing no further than 1.5 times the span of the IQR below the 25th percentile.

asynchronous under pairwise selection, 20.5% of runs resulted in cooperator fixation versus 15.7% of runs resulting in defector fixation. When updating was synchronous under pairwise selection, 59.8% of runs resulted in cooperator fixation versus roughly 26% of runs resulting in defector fixation. Across all combinations of transition rules and update dynamics, the proportion of cooperators at $G = 10000$ in the lattice model more frequently took higher values when the number of time slots was high (e.g., $t \geq 5$). As indicated in Fig. 6, under fitness proportional selection (Fig. 6a, b), the population rarely fails to reach cooperator fixation when $t > 4$, regardless of the update dynamics. Under pairwise comparison, the distribution of cooperators' final proportion of the population shifts to higher values as the number of time slots increases (Fig. 6c, d), but only under synchronous updating does it transition into a frequent state of cooperator fixation when the number of time slots is high (Fig. 6d). These results suggest that transition rules and update dynamics affect the *degree* of influence that time slots have on cooperation's growth. However, they also indicate that those modeling features do not drive cooperators' growth entirely. Instead, time slots continue to spur the *growth* of cooperators in the population, regardless of the transition rules and update dynamics employed in the simulation.

In the small-world network model, we observed a comparable pattern of findings in our sensitivity analysis. We set the probability of rewiring in our sensitivity analysis of the small-world network model to 0.05 in all simulation runs because that parameter had little effect in our initial baseline runs while leaving it fixed facilitated manageable runtimes. In these new simulation runs, we found that 71.7% of all runs in our baseline condition involving fitness proportional selection and asynchronous updating ended with the fixation of cooperators, while only 3.1% of all runs ended with fixation of defectors; a sizable percent

of runs (25.2%) failed to converge (Fig. 5e). Under fitness proportional selection and synchronous updating (Fig. 5f), the percent of runs in which cooperators reached fixation was greater (75.8%) than in our baseline condition, but so too was the percent of runs in which defector fixation occurred (21.4%); only 2.9% of runs failed to converge. Failure to achieve convergence happened regularly under pairwise selection. Roughly 81% of runs failed to converge to fixation under pairwise selection and asynchronous updating (Fig. 5g); in the instances in which convergence occurred in those runs, the population gravitated toward a state of cooperator fixation a slightly smaller percent of runs than it resulted in defector fixation (~9% versus 10% of runs). About 60.6% of runs failed to reach fixation under pairwise selection and synchronous updating (Fig. 5h), though the percent of runs resulting in the fixation of cooperators was over double the percent of runs resulting in the fixation of defectors (approximately, 26.7% versus 12.6% of runs). In the small-world network model, we also observed higher proportions of cooperators at $G = 10000$ regardless of the transition rule and update dynamics in runs that included a greater number of timeslots (Fig. 6e–h). Under fitness proportional selection (Fig. 6e, f), the population reached fixation of cooperators more often than not in the presence of multiple timeslots whether updating was asynchronous or synchronous. Under pairwise comparison, the final proportion of cooperators at $G = 10000$ more frequently exceeded the final proportion of defectors whenever $t > 3$. Also, the median proportion of cooperators at $G = 10000$ appeared to grow, albeit at a decreasing rate, with the number of time slots (Fig. 6g, h). Panels (g) and (h) furthermore indicate that the final proportion of cooperators in the population exceeded the final proportion of defectors more often than not in the instances in which convergence failed in the small-world network modeled. These findings in the small-world network model also showed

that transition rules and update dynamics affect the degree to which time slots influence the growth of cooperation, but those model features do not impede the efficacy of time slots entirely. Selection for cooperation occurs across all of our models and it does so especially when organisms can implement behavior in a relatively large number of time slots.

## Discussion

Our simulations indicate that selection for time of behavioral implementation can spur the evolution of cooperation in the spatial PD. In our baseline model, when agents can interact at more than one point in time ($t > 1$), we find that cooperators grow to fixation frequently across a wide range of parameter settings, save for when $b = 1$. Moreover, when we alter the transition rule and update dynamics of our simulation, we still find selection for cooperation and this cooperation grows more prevalent when a larger number of time slots are available for social interaction. Together, these findings yield insight into a new way that time[14,15] can influence social evolution, while also reinforcing research indicating that discrete groups[53] and network temporality[14], respectively, expand the conditions in which cooperation can evolve.

Our study also suggests new ways to extend existing models. For instance, models of movement could include intra-generational encounters that occur at various times. That is, when organisms move independent of each other, their movement could cause them to collide at different points in time, thus blending temporal and spatial assortment in substantively meaningful ways worthy of investigation.

Also, the framework used in this paper could be extended to consider more nuanced temporal strategies: instead of specifying strategies as a tuple consisting of an agent's choice in the PD and its selection of the single time at which to implement its PD behavior, strategies could consist of a PD choice and a *list* of time slots at which to play the PD. For instance, periodic behavior (implementing behavior in every other time slot, for example), punch-in-punch-out (implementing behavior in multiple adjacent time slots, but not in others), and random schedules (selecting a number of time slots randomly) could be explored. These variants, moreover, could be studied while considering behavioral types that cooperate or defect variously at different time slots. In sum, the present model can be extended in many ways.

Analysis also might focus on deriving theoretical conditions in which time slots promote cooperation, as in research that exogenously adds temporality to spatial networks[14]. The present study used computer simulation to assess the viability of selection on time as a mechanism for the evolution of cooperation; it did not offer a rigorous theoretical analysis of the properties that allow time of behavioral implementation to promote cooperation's growth. We hope future research provides that analysis.

Despite these promising avenues for inquiry, researchers should recognize the limits of our study and future ways to scrutinize those limits. For one, some might contend that we simply study a type of green beard mechanism[25,26] here. At the individual level, we would dispute that claim because the agents in our model cannot discriminate between social partners. However, at the level of the population, we observe a dynamic by which the assortment process can extinguish the presence of defectors in a given time and place, thus giving the appearance of partner discrimination. Also, we make the strong claim that temporally adjacent organisms do not interact like spatially adjacent organisms do. This assumption should be relaxed in future research such that being spatially adjacent and in a neighboring time slot might "wake up" an organism that is not implementing behavior in the same time slot.

Considering ways to enhance the verisimilitude of our model will improve insights into how cooperation can emerge from organisms' decisions about the time at which they implement social behavior. For now, we have shown in a simple model that time, a fundamental feature of existence, can enable an assortment process that allows cooperation to evolve.

## Methods

The study's models simulated the evolution of a population of $N$ organisms arranged spatially on each of two structures: a regular lattice and a small-world network. In the regular lattice, individuals shared edges/connections with $k = 4$ other organisms. In the small-world network, the parameter $k$ determined the average number of connections. At each location, organisms chose a time slot, $t$, when they would interact in the PD and this influenced the partners with whom organisms interacted—only organisms sharing an edge/link and choosing the same time slot would interact with each other.

Interactions took the form of play in a one-shot prisoner's dilemma (PD) game. In the PD, organisms could cooperate or defect. Choosing to defect when a partner chose to cooperate resulted in the payoff $b$ (a.k.a. free-riding). Joint cooperation earned $b$-$c$, whereas joint defection earned 0 and suffering free-riding resulted in –$c$. Organisms adopted strategies that formed a tuple consisting of a behavior in the PD (always cooperate or always defect) and a choice of the time slot at which to implement the PD decision.

Payoffs from the implementation of these strategies influenced the rate at which strategies were adopted according to a combination of transition rules and update dynamics[5,8]. The transition rules consisted of death-birth fitness proportional selection (hereafter, "fitness proportional selection") and pairwise comparison. Under fitness proportional selection, payoffs were tallied after organisms played the PD; then, one or more organisms were selected at random to die at the end of the current generation, with the organism(s) replacing the dead organism(s) adopting one of the model's strategies with a probability proportional to each strategy's payoffs. When using asynchronous updating, only one organism died and was replaced; when using synchronous updating, all agents die and are replaced. Under asynchronous pairwise comparison, we randomly identify an organism in the population (the "focal" organism) and, then, randomly select a neighbor of that organism; if the neighbor has higher payoffs, then the focal organism adopts the neighbor's attributes. Under synchronous pairwise comparison, we repeat the aforementioned pairwise comparison process for all members of the population—i.e. making all comparisons before replacement—and, then, replace the population in one batch process.

We studied the population subject to one combination of parameter settings, transition rules, and update dynamics for 10000 generations, which we label one "run" of the simulation. For every combination of parameters in the model, we ran the model for 7 runs.

The study exogenously varied model parameters to understand whether and how strategy evolution changed according to the values those parameters took. We drew $N$ from $\mathbf{N} = \{100, 225, 400\}$, $t$ from $\mathbf{T} = \{1, 2, …, 10\}$, and $b$ from $\mathbf{B} = \{1, 2, …, 10\}$ (note that $c = 1$ in all simulations). The parameter $k$ was fixed at 4 in the regular lattice, but in the small-world network model, we drew $k$ from $\mathbf{K} = \{2, 3, 4\}$. The small-world network model was studied under two re-wiring probabilities, 0.05 and 0.15, in our baseline condition of fitness proportional selection and asynchronous updating. Then, to achieve reasonable runtimes, we set the parameter equal to 0.05 for all other combinations of update dynamic and transition rules due to the virtually null effect of the parameter on results in the baseline model. The simulation explored the full parameter space and did so, as mentioned above, in 7 replications (i.e. runs) for each parameter combination.

**Reporting summary**. Further information on research design is available in the Nature Research Reporting Summary linked to this article.

## Code availability

Simulations were written in Python 3.7.6. Data generated in the simulations were analyzed in R 3.5.3. All computer code is available online via a project page hosted by the Open Science Framework (https://doi.org/10.17605/OSF.IO/3JSXV)[56].

## Data availability

Data sets used in the study are publicly available online via a project page hosted by the Open Science Framework (https://doi.org/10.17605/OSF.IO/3JSXV)[56]. Computer code provided at that same project page can be used to reproduce all figures in this paper using those data.

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

## Author contributions

T.J. came up with the idea; T.J. and O.S. planned the research; O.S. programmed the simulations, with T.J. validating and double-checking them; T.J. performed the data analysis, with O.S. double-checking it; T.J. wrote the first draft of the paper and O.S. provided substantial revisions and amendments in subsequent drafts.

## Competing interests

The authors declare no competing interests.
