## [Transparent Peer Review File · Communications Biology]

Reviewers' comments:

Reviewer #1 (Remarks to the Author):

Starting from the evolution of cooperation on networks, here authors introduce time slots for individuals. And then individuals are only allowed to interact with spatially adjacent neighbors who are also at their same time slot. They find that cooperation could persist even when the simple rule $b/c > k$ is missing. Although simulations are performed on both homogeneous and heterogeneous networks, the authors may wish to consider the following criticisms in their revisions.

The authors mention some scenarios pertaining to social cooperative behavior. While I cannot find the motivation supporting the introduction of the time slot. What exactly constitutes an interaction in each of these cases, and why should this type of interaction tell readers about the evolution of cooperation. Are they playing prisoner dilemma when people "meet with neighbors on weekend mornings to clean-up litter in their surroundings"?

Only an individual is given the chance to update its strategy in each generation. This confers the slow asynchronous update. I was wondering whether the results would change after the synchronous update, for example, pairwise comparison in (Traulsen, A., Nowak, M. A. & Pacheco, J. M. Stochastic dynamics of invasion and fixation. Phys. Rev. E 74, 011909 (2006).) It is not permissible by just looking at a single choice of microscopic process unless a clear argument is provided (which is not the case here).

That is far from enough to only consider the case where 75% cooperators exist for scale-free networks, where defectors still have a high chance to take over (re-invade) the whole population. I guess synchronous update could solve this dilemma in some sense (like pair approximation). But authors should at least consider the case where the absorbing state is reached for sure even with a small population size. Indeed, the two absorbing states in the stochastic process must be reached before drawing any conclusive result.

This conclusion "northeast corner of the grids indicates the increased success of cooperation when both b and t take large values." presented in Fig. 3 is uncertain. For $N=100$, the authors only consider t from 1 to 10 to draw the above conclusion from simulations. It is apparent that they are wrong by looking at $t > 100$ where individuals are all at different time slots, and no one would interact, therefore there are only half cooperators from the initial condition. I suggest the authors think about theoretical insights, which are all absent for many results in the manuscript.

Minor: The Introduction section is too long to get the authors' contribution/invention easily.

Reviewer #2 (Remarks to the Author):

Johnson and Smirnov present a simulation study of the one-shot Prisoner's Dilemma played on a graph, in which interactions are constrained both by the graph structure and by the choice of time slot in which a player is active. This time slot choice is assumed to be a feature of a player's strategy, and thus both behavior (cooperate or defect) and interaction partner (who a player shares a timeslot with) coevolve, although the latter is constrained by the number of available time slots and the player's position on the graph.

The idea of varying the timing of social interactions – the authors give the example of two organisms living in the same place, but one being nocturnal and the other diurnal, to illustrate the motivation for the model – makes a lot of sense. As the authors themselves point out there is some qualitative similarity to other models that look at the effects of e.g. movement, or perhaps dispersal strategy, but

as far as I know their conceptualization of timing is novel.

The big problem with the paper is the claim that their results violate the "rule" $b/c > k$ which must be satisfied for cooperation to evolve on a graph (where k is the average number of neighbors on the graph). It is pretty clear that to analyze the evolutionary dynamics on the types of graph they study we would need to know the average number of players who are neighbors in the same time slot. That is, there is some effective k that accounts for both time and space. Any revised version of the article must account for this, e.g. by presenting a figure showing the effective k and how it compares to b/c . This is important because claiming that the rule $b/c > k$ does not hold could cause enormous confusion in the literature if not properly contextualized. It is quite reasonable to point out that graph structure alone cannot tell us if cooperation will evolve if we do not know the timing of interactions, but if that is the case it is because the graph structure alone does not tell us the average number of interactions between players.

Point-by-Point Response to Reviewers' Comments on "Temporal assortment of cooperators in the spatial prisoner's dilemma"

Response to Reviewer #1 Report

Reviewer #1 begins with a crisp summary of our manuscript:

Starting from the evolution of cooperation on networks, here authors introduce time slots for individuals. And then individuals are only allowed to interact with spatially adjacent neighbors who are also at their same time slot. They find that cooperation could persist even when the simple rule $b/c > k$ is missing. Although simulations are performed on both homogeneous and heterogeneous networks, the authors may wish to consider the following criticisms in their revisions.

(Reviewer 1, Paragraph 1)

Next, Reviewer 1 notes that we inadequately motivate our study:

The authors mention some scenarios pertaining to social cooperative behavior. While I cannot find the motivation supporting the introduction of the time slot. What exactly constitutes an interaction in each of these cases, and why should this type of interaction tell readers about the evolution of cooperation. Are they playing prisoner dilemma when people "meet with neighbors on weekend mornings to clean-up litter in their surroundings"?

(Reviewer 1, Paragraph 2)

This second paragraph of Reviewer 1's report draws attention to a problem that we, too, noticed after spending some time away from our manuscript during the review process. The examples in the original manuscript (a) fail to highlight that the social interactions occur at a given point in time even though they could occur at other moments in time (i.e. the problem raised in sentences 1 and 2 of Reviewer 1's second paragraph) and (b) inadequately explain that each scenario involves a prisoner's dilemma (i.e. the problem raised in sentences 3 and 4 of Reviewer 1's second paragraph). Accordingly, we have revised the main text to include two detailed examples that highlight both the role of time slots and the PD incentive structure. Specifically, we write:

Cooperation not only occurs at specific locations, but, also, it takes place at particular points in time. Indeed, whenever organisms interact in the flesh, cooperation necessarily occurs at both a spatial location and temporal moment.

For instance, at night^{1,2}, spinner dolphins collectively herd prey, creating food aggregations that exceed the prey densities encountered by lone foragers³. After amassing their prey, the encircling herders enter the accumulation in pairs and take turns feeding³. Each pair feeds for roughly the same duration, thus resisting the opportunity to enjoy a disproportionate individual gain that would diminish others' benefits³. Also, only participants in the herding appear to partake in the feeding³, even though, conceivably, non-herders could lurk nearby and attempt to consume the accumulated prey. This herding behavior amounts to cooperation in a prisoner's dilemma: the herders create a benefit, b , at a cost, c , that free riders could consume without paying a cost. The behavior's subtleties, however, obscure a seemingly mundane feature of it—namely, that it occurs at one point in time (a particular moment at night), as opposed to other points in time.

The temporal concentration of cooperation also occurs in other species. Consider humans engaged in illicit market transactions that occur beyond the scope of institutions that support cooperative trade. These traders aim to acquire a benefit, b , by exchanging a resource that they part with at personal cost, c . However, they do best when their trade partner delivers b , while they transfer poor-quality goods or inadequate currency that reduce or nullify c . To avoid

such cheating, illegal barterers schedule the time of transactions deliberately, or express preferences for some hours over others⁴. Cooperation, these activities imply, succeeds more frequently at certain times than others.

(Revised Manuscript, Paragraphs 1-3)

We believe this revision introduces the reader to the prisoner's dilemma more clearly and, then, it underscores that the time at which the interaction occurs need not be taken as given. In recognition of Reviewer 2's view that "The idea of varying the timing of social interactions...makes a lot of sense" (Reviewer 2, paragraph 2), we have aimed not to belabor the examples, thus striking a balance between the comments of Reviewer 1 and Reviewer 2. We thank Reviewer 1 for encouraging us to make this improvement to our manuscript and we appreciate Reviewer 2's confidence in the intuition underlying our model's main feature.

Reviewer 1 next raises a question about whether our results would differ if we altered the update dynamics of our model and, then, in turn, proposes that we consider varying our update dynamics as a robustness check:

Only an individual is given the chance to update its strategy in each generation. This confers the slow asynchronous update. I was wondering whether the results would change after the synchronous update, for example, pairwise comparison in (Traulsen, A., Nowak, M. A. & Pacheco, J. M. Stochastic dynamics of invasion and fixation. *Phys. Rev. E* 74, 011909 (2006).) It is not permissible by just looking at a single choice of microscopic process unless a clear argument is provided (which is not the case here).

(Reviewer 1, Paragraph 3)

Reviewer 1 rightly points out that we needed to consider alternative update dynamics⁵ in our model as we solely studied our model using an asynchronous strategy update process and a death-birth fitness proportional selection transition rule. Reviewer 1 suggests studying our model under both an additional strategy update processes (namely, synchronous updating) and an alternative transition rule⁵ (namely, pairwise comparison). We implemented each of these suggestions. We studied the model under combinations of update processes (asynchronous and synchronous) and transition rules (death-birth fitness proportional selection and pairwise comparison). These changes resulted in lengthy simulation runtimes and multiplied the time needed to validate and analyze the simulation data, thus we narrowed our focus to two spatial structures (the lattice and small-world network models). We describe these methodological changes in our revised Methods section. Our revised Results section describes the findings of those methods, as does the revised Abstract and Introduction, as appropriate. Our revised Methods section now reads as follows:

Methods

The study's models simulated the evolution of a population of N organisms arranged spatially on each of two structures: a regular lattice and a small-world network. In the regular lattice, individuals shared edges/connections with $k=4$ other organisms. In the small-world network, the parameter k determined the average number of connections. At each location, organisms chose a time slot, t , when they would interact in the PD and this influenced the partners with whom organisms interacted—only organisms sharing an edge/link and choosing the same time slot would interact with each other.

Interactions took the form of play in a one-shot prisoner's dilemma (PD) game. In the PD, organisms could cooperate or defect. Choosing to defect when a partner chose to cooperate resulted in the payoff b (a.k.a. free-riding). Joint cooperation earned $b-c$, whereas joint defection earned 0 and suffering free-riding resulted in $-c$. Organisms adopted strategies that formed a tuple consisting of a behavior in the PD (always cooperate or always defect) and a choice of the time slot at which to implement the PD decision.

Payoffs from the implementation of these strategies influenced the rate at which strategies were adopted according to a combination of transition rules and update dynamics^{5,6}. The transition rules consisted of death-birth fitness proportional selection (hereafter, “fitness proportional selection”) and pairwise comparison. Under fitness proportional selection, payoffs were tallied after organisms played the PD; then, one or more organisms were selected at random to die at the end of the current generation, with the organism(s) replacing the dead organism(s) adopting one of the model’s strategies with a probability proportional to each strategy’s payoffs. When using asynchronous updating, only one organism died and was replaced; when using synchronous updating, all agents die and are replaced. Under asynchronous pairwise comparison, we randomly identify an organism in the population (the “focal” organism) and, then, randomly select a neighbor of that organism; if the neighbor has higher payoffs, then the focal organism adopts the neighbor’s attributes. Under synchronous pairwise comparison, we repeat the aforementioned pairwise comparison process for all members of the population—i.e. making all comparisons before replacement—and, then, replace the population in one batch process.

We studied the population subject to one combination of parameter settings, transition rules, and update dynamics for 10000 generations, which we label one “run” of the simulation. For every combination of parameters in the model, we ran the model for 7 runs.

The study exogenously varied model parameters to understand whether and how strategy evolution changed according to the values those parameters took. We drew N from $N=\{100, 225, 400\}$, t from $T=\{1, 2, \dots, 10\}$, and b from $B=\{1, 2, \dots, 10\}$ (note that $c=1$ in all simulations). The parameter k was fixed at 4 in the regular lattice, but in the small-world network model, we drew k from $K=\{2, 3, 4\}$. The small-world network model was studied under two re-wiring probabilities, 0.05 and 0.15, in our baseline condition of fitness proportional selection and asynchronous updating. Then, to achieve reasonable runtimes, we set the parameter equal to 0.05 for all other combinations of update dynamic and transition rules due to the virtually null effect of the parameter on results in the baseline model. The simulation explored the full parameter space and did so, as mentioned above, in 7 replications (i.e. runs) for each parameter combination.

Code Availability. Simulations were written in Python 3.7.6. Data generated in the simulations were analyzed in R 3.5.3. All computer code used in the simulations and data analysis is provided in the Supplementary Information.

Data Availability. Data sets used in the study are publicly available online and can be accessed via execution of the R code available in the Supplementary Information.

(Revised Manuscript, pages 9-10)

The additions to our Results section in response to the reviewers’ comments are as follows:

The frequency of cooperator fixation, however, does vary in sensitivity analyses aimed at examining the consequences of altering the transition rule and update process of our simulation⁵⁻⁸. In those sensitivity analyses, we replicated our simulation under combinations of two different transition rules (fitness proportional selection and pairwise comparison) and two separate update dynamics (synchronous and asynchronous updating), leading to a 2x2 research design for each spatial structure (see Table 1). We treat the transition rule and update dynamics deployed in the models reported above (viz. fitness proportional selection and asynchronous updating) as a baseline condition.

In the lattice model, we found that about 73% of all runs in our baseline condition ended with the fixation of cooperators, approximately 8.1% of all runs ended with fixation of defectors, and 18.9% of all runs failed to converge (Fig.5(i)). Under fitness proportional selection and synchronous updating (Fig.5(ii)), the percent of runs in which cooperators reached fixation was fewer (61.6%), though it still exceeded the percent of runs resulting in defector fixation (32.2%), and only 6.1% of runs failed to converge. Failed convergence occurred more frequently when pairwise selection served as the model’s transition rule; 63.8% of runs failed to converge when that transition rule was combined with asynchronous updating (Fig.5(iii)) and 14.3% of runs failed to converge when pairwise selection was combined with synchronous updating (Fig.5(iv)). When convergence succeeded, fixation of cooperators occurred more frequently than defector fixation

in models with pairwise comparison. When updating was asynchronous under pairwise selection, 20.5% of runs resulted in cooperator fixation versus 15.7% of runs resulting in defector fixation. When updating was synchronous under pairwise selection, 59.8% of runs resulted in cooperator fixation versus roughly 26% of runs resulting in defector fixation. Across all combinations of transition rules and update dynamics, the proportion of cooperators at $G=10000$ in the lattice model more frequently took higher values when the number of time slots was high (e.g., $t \geq 5$). As indicated in Fig.6, under fitness proportional selection (Fig.6(i)-(ii)), the population rarely fails to reach cooperator fixation when $t > 4$, regardless of the update dynamics. Under pairwise comparison, the distribution of cooperators' final proportion of the population shifts to higher values as the number of time slots increases (Fig.6(iii)-(iv)), but only under synchronous updating does it transition into a frequent state of cooperator fixation when the number of time slots is high (Fig.6(iv)). These results suggest that transition rules and update dynamics affect the degree of influence that time slots have on cooperation's growth. However, they also indicate that those modeling features do not drive cooperators' growth entirely. Instead, time slots continue to spur the growth of cooperators in the population, regardless of the transition rules and update dynamics employed in the simulation.

In the small-world network model, we observed a comparable pattern of findings in our sensitivity analysis. We set the probability of rewiring in our sensitivity analysis of the small-world network model to 0.05 in all simulation runs because that parameter had little effect in our initial baseline runs while leaving it fixed facilitated manageable runtimes. In these new simulation runs, we found that 71.7% of all runs in our baseline condition involving fitness proportional selection and asynchronous updating ended with the fixation of cooperators, while only 3.1% of all runs ended with fixation of defectors; a sizable percent of runs (25.2%) failed to converge (Fig.5(v)). Under fitness proportional selection and synchronous updating (Fig.5(vi)), the percent of runs in which cooperators reached fixation was greater (75.8%) than in our baseline condition, but so too was the percent of runs in which defector fixation occurred (21.4%); only 2.9% of runs failed to converge. Failure to achieve convergence happened regularly under pairwise selection. Roughly 81% of runs failed to converge to fixation under pairwise selection and asynchronous updating (Fig.5(vii)); in the instances in which convergence occurred in those runs, the population gravitated toward a state of cooperator fixation a slightly smaller percent of runs than it resulted in defector fixation (approximately 9% versus 10% of runs). About 60.6% of runs failed to reach fixation under pairwise selection and synchronous updating (Fig.5(viii)), though the percent of runs resulting in the fixation of cooperators was over double the percent of runs resulting in the fixation of defectors (approximately, 26.7% versus 12.6% of runs). In the small-world network model, we also observed higher proportions of cooperators at $G=10000$ regardless of the transition rule and update dynamics in runs that included a greater number of timeslots (Fig.6(v)-(viii)). Under fitness proportional selection (Fig.6(v)-(vi)), the population reached fixation of cooperators more often than not in the presence of multiple timeslots whether updating was asynchronous or synchronous. Under pairwise comparison, the final proportion of cooperators at $G=10000$ more frequently exceeded the final proportion of defectors whenever $t > 3$. Also, the median proportion of cooperators at $G=10000$ appeared to grow, albeit at a decreasing rate, with the number of time slots (Fig.6(vii)-(viii)). Panels (vii) and (viii) furthermore indicate that the final proportion of cooperators in the population exceeded the final proportion of defectors more often than not in the instances in which convergence failed in the small-world network modeled. These findings in the small-world network model also showed that transition rules and update dynamics affect the degree to which time slots influence the growth of cooperation, but those model features do not impede the efficacy of time slots entirely. Selection for cooperation occurs across all of our models and it does so especially when organisms can implement behavior in a relatively large number of time slots.

(Revised Manuscript, pages 6-7)

By performing sensitivity analyses that considered the influence of update dynamics and transition rules on our findings, we have provided a more thorough investigation of the robustness of our results. We appreciate Reviewer 1 for encouraging us to take such measures as they have enhanced an understanding of both the durability and limitations of our findings.

In a similar spirit, Reviewer 1 disputed our use of a low-threshold for model convergence:

That is far from enough to only consider the case where 75% cooperators exist for scale-free networks, where defectors still have a high chance to take over (re-invade) the whole population. I guess synchronous update could solve this dilemma in some sense (like pair approximation). But authors should at least consider the case where the absorbing state is reached for sure even with a small population size. Indeed, the two absorbing states in the stochastic process must be reached before drawing any conclusive result.
(Reviewer 1, Paragraph 4)

Reviewer 1 is right to have criticized this methodological decision and we have adopted a strict definition of convergence (100% of the population adopting a strategy) in our revised manuscript. To facilitate the reader's understanding of our results when the models do not converge, we report the final proportion of cooperators in the population across simulation runs in our revised manuscript; however, we make no pretenses that these final population shares reflect an absorbing state of the simulation. Instead, they represent a snapshot of a simulation in its final generation and we believe that the manuscript does not imply that such information should be interpreted as evidence that cooperation has reached fixation. Because this approach to presenting our results appears throughout the Results section and figures, we refrain from copying and pasting the entirety of those sections here; instead, we request that reviewers consult those sections of the main text.

Next, Reviewer 1 notes an important caveat about how we describe our findings.

This conclusion "northeast corner of the grids indicates the increased success of cooperation when both b and t take large values." presented in Fig. 3 is uncertain. For $N=100$, the authors only consider t from 1 to 10 to draw the above conclusion from simulations. It is apparent that they are wrong by looking at $t>100$ where individuals are all at different time slots, and no one would interact, therefore there are only half cooperators from the initial condition. I suggest the authors think about theoretical insights, which are all absent for many results in the manuscript.

(Reviewer 1, Paragraph 5)

Reviewer 1 is correct that our phrasing in that portion of the manuscript fails to account for the theoretical observation that the size of the population puts an upper limit on the number of time slots that can lead to viable cooperation. To ensure that readers recognize this possibility and the theoretical limitations of our investigation, we have taken several measures. First, we have changed the phrasing of the sentence that Reviewer 1 excerpts. We now point out that the statement only pertains to the range of parameter values we consider. Specifically, we write, "*the prevalence of light blue tiles in the northeast corner of the grids indicates the increased success of cooperation when both b and t take large values within the model's parameter range*" (p.16). Second, we have added sentences in the Results section that explain the reasoning that Reviewer 1 accurately conveys and which allude to the need for subsequent, rigorous analysis in future research:

However, the increasing likelihood of cooperator fixation with growth in the number of time slots clearly has theoretical limits as astutely pointed out by an anonymous reviewer of this article. Were $t \rightarrow N$ or exceed it, the population would become sparsely distributed across time slots, leading inevitably to an asocial state in which cooperation would fail. In the discussion section, we propose analytic approaches to gain insight into such possibilities, though we preclude such phenomena by our choice of parameters in the current model.

(Revised Manuscript, page 4)

Third, we acknowledge in the discussion section that the present paper only offers simulation results that assess the viability of time as a mechanism for cooperation's evolution and it stops short of a genuine theoretical analysis. After discussing ways to extend our work in the discussion section, we write:

Analysis also might focus on deriving theoretical conditions in which time slots promote cooperation. The present study used computer simulation to assess the viability of time as a mechanism for the evolution of cooperation; it did not offer a rigorous theoretical analysis of the properties that allow time of behavioral implementation to promote cooperation's growth. We hope future research provides that analysis.

(Revised Manuscript, page 8)

We believe this addition to our discussion section conveys the limits of our current investigation accurately. Subsequent theoretical work uncovering the precise conditions in which time slots promote cooperation will be valuable and we contend that our current work paves the way for such research by proposing a novel mechanism that, in simulations, fosters cooperation's evolution.

Finally, Reviewer 1 notes that our introduction is too long and we need to present our innovation to the literature sooner in the opening section:

Minor: The Introduction section is too long to get the authors' contribution/invention easily.

(Reviewer 1, Paragraph 6)

Upon re-reading our manuscript, we agree with Reviewer 1's point that we need to highlight our contribution sooner in the introduction. As a result, we have taken the final two paragraphs of the introduction, which describe our model in detail, and we now place them immediately after the motivating examples that appear at the start of the introduction (see page 2 of our revised manuscript). This mode of presentation will help the reader discern the novel features of our model early in the manuscript. However, to ensure that all readers understand the distinctiveness and scope of our contribution, we have retained the paragraphs that distinguish our research from work that is both substantively and superficially similar. As a result, the introduction section remains similar in length, but we believe that it is imperative to provide the reader an adequate understanding of where our study fits in the literature. We believe it would be too easy for some readers to conflate our research with prior studies or to ignore how our study relates to previous research, thus we have attempted to strike a balance with our revision. We now prioritize the description of our model (by placing it in the paper's second paragraph) to ensure that our contribution/invention can be discerned early in the introduction, but we also retain the literature review to make sure that readers receive a full explanation of how our work relates to previous work on the topic. Furthermore, in compliance with the style requirements of *Communications Biology*, we have added a final paragraph to our introduction that summarizes our findings.

In general, we believe all of the above changes have improved our manuscript. We thank Reviewer 1 for providing the comments that inspired these changes.

Response to Reviewer #2 Report

Reviewer 2 begins with an elegant summary of our research (Reviewer 2, paragraph 1), as well as an assessment of the intuitiveness and novelty of the mechanism studied in our research:

Johnson and Smirnov present a simulation study of the one-shot Prisoner's Dilemma played on a graph, in which interactions are constrained both by the graph structure and by the choice of time slot in which a player is active. This time slot choice is assumed to be a feature of a player's strategy, and thus both behavior (cooperate or defect) and interaction partner (who a player shares a timeslot with) coevolve, although the latter is constrained by the number of available time slots and the player's position on the graph.

The idea of varying the timing of social interactions – the authors give the example of two organisms living in the same place, but one being nocturnal and the other diurnal, to illustrate the motivation for the model – makes a lot of sense. As the authors themselves point out there is some qualitative similarity to other models that look at the effects of e.g. movement, or perhaps dispersal strategy, but as far as I know their conceptualization of timing is novel.

(Reviewer 2, Paragraphs 1-2)

After providing those opening remarks, Reviewer 2 rightfully points out a problem with the first version of our paper:

The big problem with the paper is the claim that their results violate the “rule” $b/c > k$ which must be satisfied for cooperation to evolve on a graph (where k is the average number of neighbors on the graph). It is pretty clear that to analyze the evolutionary dynamics on the types of graph they study we would need to know the average number of players who are neighbors in the same time slot. That is, there is some effective k that accounts for both time and space. Any revised version of the article must account for this, e.g. by presenting a figure showing the effective k and how it compares to b/c . This is important because claiming that the rule $b/c > k$ does not hold could cause enormous confusion in the literature if not properly contextualized. It is quite reasonable to point out that graph structure alone cannot tell us if cooperation will evolve if we do not know the timing of interactions, but if that is the case it is because the graph structure alone does not tell us the average number of interactions between players.

(Reviewer 2, Paragraph 3)

In retrospect, we regret how we phrased this feature of our findings. We fully agree with Reviewer 2's point and we appreciate Reviewer 2's description of how to think about our findings relative to the condition that Ohtsuki et al. identified. In order to address this issue, we have removed discussion of this feature of the findings from the abstract and we have revised the corresponding discussion in the Results section. That section now acknowledges Reviewer 2's insightful observation and reads:

At first glance, Fig. 3 seems to suggest that the findings violate Ohtsuki et al.'s rule for the evolution of cooperation on graphs, which stipulates that selection favors cooperation when the ratio of benefits to costs exceeds k , the average number of neighboring organisms⁹. In 54.8% of all runs, cooperators grow to fixation when $b/c \not> k$ (i.e. when $b \leq 4$, given $c=1$ and $k=4$) and multiple time slots are present ($t \geq 2$). This apparent violation of the canonical rule $b/c > k$, however, is superficial; as another anonymous reviewer adroitly recognized, organisms experience an effective k , which we label k^ , denoting the average neighbors at a given temporal-spatial location. Were organisms uniformly distributed across spatial locations and time slots, k^* would equal k/t . However, random seeding of time slots and the evolution of game play in those slots distributes the population unevenly; thus, $k^* = k/t$ constitutes the minimum value that*

parameter can take. This equation indicates that the addition of time slots has the effect of decreasing the value of k —a phenomenon observed in other recent research¹⁰. When $t > 1$, k^* is less than k and one would expect cooperation to evolve when $b/c > k^*$, thus explaining why apparent violations of the rule $b/c > k$ appear in the data. In the discussion section, we address the significant implications this observation holds for empirical studies that seek to test the rule, $b/c > k$, in conditions where behavior varies across both space and time.

(Revised Manuscript, page 5)

We believe that this revised discussion now clarifies that the apparent violation of Ohtsuki et al.'s condition for the evolution of cooperation on graphs is superficial and we note that other recent models¹⁰ have observed the same phenomenon in a different context (namely, in the context of game transitions as a mechanism for cooperation's evolution). Because this phenomenon has been previously identified (i.e. that certain mechanisms can effectively change the value of k in the model), we elected not to focus undue attention on that feature of our results. Doing so seemed to imbue the finding with excess significance and it gave the appearance of novelty when, in fact, the research cited previously in this paragraph already noticed the phenomenon we observe, albeit in relation to a different mechanism. We believe that the above discussion of the issue provides appropriate clarification of the matter, while not placing undue weight on it. We thank Reviewer 2 for drawing our attention to this area for improvement; it has enhanced our paper considerably.

Conclusion of Point-by-Point Response

Reviewers 1 and 2 offered valuable comments that have helped us improve our paper. We have attempted to address each comment offered by the reviewers and, in so doing, we believe that we have produced a much better paper. Both reviewers have our gratitude.

References Cited in this Response

- 1 Norris, K. S. & Dohl, T. P. Behavior of the Hawaiian spinner dolphin, *Stenella Longirostris*. *Fishery Bulletin* **77**, 821-849 (1980).
- 2 Benoit-Bird, K. J. & Au, W. W. L. Prey dynamics affect foraging by a pelagic predator (*Stenella longirostris*) over a range of spatial and temporal scales. *Behavioral Ecology and Sociobiology* **53**, 364-373, doi:10.1007/s00265-003-0585-4 (2003).
- 3 Benoit-Bird, K. J. & Au, W. W. L. Cooperative prey herding by the pelagic dolphin, *Stenella longirostris*. *Journal of the Acoustical Society of America* **125**, 125-137 (2009).
- 4 Jacobs, B. A. *Dealing crack: the social world of streetcorner selling*. (Northeastern University Press, 1999).
- 5 Grilo, C. & Correia, L. The Influence of the Update Dynamics on the Evolution of Cooperation. *International Journal of Computational Intelligence Systems* **2**, 104-114, doi:10.1080/18756891.2009.9727645 (2009).
- 6 Wu, B., Bauer, B., Galla, T. & Traulsen, A. Fitness-based models and pairwise comparison models of evolutionary games are typically different—even in unstructured populations. *New Journal of Physics* **17**, 023043, doi:<http://dx.doi.org/10.1088/1367-2630/17/2/023043> (2015).
- 7 Ohtsuki, H. & Nowak, M. A. The replicator equation on graphs. *Journal of Theoretical Biology* **243**, 86-97, doi:<https://doi.org/10.1016/j.jtbi.2006.06.004> (2006).

- 8 Liu, X., He, M., Kang, Y. & Pan, Q. Fixation of strategies with the Moran and Fermi processes in evolutionary games. *Physica A: Statistical Mechanics and its Applications* **484**, 336-344, doi:<https://doi.org/10.1016/j.physa.2017.04.154> (2017).
- 9 Ohtsuki, H., Hauert, C., Lieberman, E. & Nowak, M. A. A simple rule for the evolution of cooperation on graphs and social networks. *Nature* **441**, 502-505, doi:http://www.nature.com/nature/journal/v441/n7092/supinfo/nature04605_S1.html (2006).
- 10 Su, Q., McAvoy, A., Wang, L. & Nowak, M. A. Evolutionary dynamics with game transitions. *Proceedings of the National Academy of Sciences* **116**, 25398-25404, doi:10.1073/pnas.1908936116 (2019).

REVIEWERS' COMMENTS:

Reviewer #1 (Remarks to the Author):

The authors have fully addressed my comments in a wonderful way, and would thus like to recommend the acceptance.

Reviewer #2 (Remarks to the Author):

The authors have done a really good job of addressing my concerns and I now feel that the manuscript represents a valuable contribution. I support publication.

**Response to Referee's Final Round Comments on
"Temporal assortment of cooperators in the spatial prisoner's dilemma"**

Tim Johnson and Oleg Smirnov

To comply with the file checklist provided by *Communications Biology*, we have created this document to provide a response to the referee's comments on the most-recent version of our manuscript. The referee's comments on that manuscript were as follows:

Reviewer #1 (Remarks to the Author):

The authors have fully addressed my comments in a wonderful way, and would thus like to recommend the acceptance.

Reviewer #2 (Remarks to the Author):

The authors have done a really good job of addressing my concerns and I now feel that the manuscript represents a valuable contribution. I support publication.

Given these positive comments, we have not made any substantive changes to our manuscript; we have only altered the formatting of the manuscript to comply with the guidance provided in both the Editorial Request document and the email sent to us on September 14, 2021 by Associate Editor Dr. Caitlin Karniski.

We thank the reviewers and the editors of our manuscript for their insightful comments throughout the review process. Those suggestions improved our paper considerably.